# Artificial Small RNA-Based Silencing Tools for Antiviral Resistance in Plants

**DOI:** 10.3390/plants9060669

**Published:** 2020-05-26

**Authors:** Adriana E. Cisneros, Alberto Carbonell

**Affiliations:** Instituto de Biología Molecular y Celular de Plantas, Consejo Superior de Investigaciones Científicas-Universitat Politècnica de València, 46022 Valencia, Spain; adgoncis@ibmcp.upv.es

**Keywords:** RNA silencing, artificial small RNA, amiRNA, atasiRNA, syn-tasiRNA, antiviral resistance, VSR, plant virus, viroid

## Abstract

Artificial small RNAs (art-sRNAs), such as artificial microRNAs (amiRNAs) and synthetic trans-acting small interfering RNAs (syn-tasiRNAs), are highly specific 21-nucleotide small RNAs designed to recognize and silence complementary target RNAs. Art-sRNAs are extensively used in gene function studies or for improving crops, particularly to protect plants against viruses. Typically, antiviral art-sRNAs are computationally designed to target one or multiple sites in viral RNAs with high specificity, and art-sRNA constructs are generated and introduced into plants that are subsequently challenged with the target virus(es). Numerous studies have reported the successful application of art-sRNAs to induce resistance against a large number of RNA and DNA viruses in model and crop species. However, the application of art-sRNAs as an antiviral tool has limitations, such as the difficulty to predict the efficacy of a particular art-sRNA or the emergence of virus variants with mutated target sites escaping to art-sRNA-mediated degradation. Here, we review the different classes, features, and uses of art-sRNA-based tools to induce antiviral resistance in plants. We also provide strategies for the rational design of antiviral art-sRNAs and discuss the latest advances in developing art-sRNA-based methodologies for enhanced resistance to plant viruses.

## 1. Introduction

RNA interference (RNAi) is a biological process conserved in most eukaryotes and characterized by the sequence-specific degradation of target RNA by complementary small RNAs (sRNAs) [1]. RNAi pathways are triggered by double-stranded RNA (dsRNA) processed into sRNA duplexes by Dicer ribonucleases [1,2]. One of the strands of the duplex is preferentially loaded into an ARGONAUTE (AGO) protein, and the resulting complex, termed RNA-induced silencing complex (RISC), recognizes and silences complementary target RNA through diverse mechanisms [3,4]. The seminal observation that exogenously applied dsRNA can artificially trigger RNAi of specific genes in *Caenorabtitis elegans* [1] fueled the development of a plethora of RNAi-based tools in multiple organisms for the study of gene function and for more applied purposes, including medical therapies and diverse biotechnological uses.

In plants, RNAi tools have been extensively used to confer antiviral resistance. Early RNAi approaches, such as virus-induced gene silencing (VIGS) and hairpin (hp)-based silencing, consisted of the expression of dsRNA or hpRNA precursors, respectively, bearing sequences of the target virus (for a recent review see [5]). Although very popular, these approaches lacked high specificity as the large populations of sRNAs produced from these type of precursors favor the accidental targeting of complementary cellular transcripts [6]. This limitation was overcome with the development of a series of tools based on artificial sRNAs (art-sRNAs) [7], 21-nucleotide sRNAs expressed in planta from endogenous sRNA precursors and computationally designed to silence target RNAs with high specificity [8].

Here, we describe the different classes, features, and uses of art-sRNA-based RNAi (art-sRNAi) tools to target viral RNAs and induce antiviral resistance in plants. We also provide basic rules for art-sRNA design and describe which viral regions are typically targeted. Finally, we discuss the latest art-sRNAi strategies providing enhanced antiviral resistance, such as the high-throughput identification of highly effective antiviral art-sRNAs and their simultaneous co-expression for the multi-targeting of viral RNAs.

## 2. Classes, Features, and Uses of Antiviral Art-sRNAi Tools

Artificial microRNAs (amiRNAs) and artificial/synthetic trans-acting small interfering RNAs (atasiRNAs/syn-tasiRNAs, hereafter syn-tasiRNAs) are the two main classes of plant art-sRNAs [8]. Despite differing in their biogenesis pathway (see below), both classes of art-sRNAs function similarly by associating with AGO1 to specifically silence target RNAs through endonucleolytic cleavage or translation repression (Figure 1).

### 2.1. AmiRNAs

AmiRNAs are produced in planta by expressing amiRNA transgenes containing a functional plant *MIRNA* precursor in which the endogenous miRNA guide/miRNA star sequences are substituted by the amiRNA guide/amiRNA star sequences (Figure 1a) [8,9]. Importantly, other sequences of the precursor are modified to preserve the original secondary structure required for accurate Dicer-like 1 (DCL1) processing. The amiRNA transgene is transcribed to produce the amiRNA primary transcript (pri-amiRNA). Sequential processing by DCL1 produces the pre-amiRNA followed by the amiRNA duplex, and the amiRNA guide strand is incorporated into AGO1 to target complementary viral RNA typically in a single site (Figure 1a).

AmiRNAs were first used to confer single and dual resistance against two RNA viruses, *Turnip yellow mosaic virus* (TYMV) and *Turnip mosaic virus* (TuMV), in transgenic *Arabidopsis thaliana* (Arabidopsis) plants expressing one or two amiRNA transgenes, respectively [10]. In this pioneering study, amiRNAs were designed to target a single site in P69 and HC-Pro RNAs of TYMV and TuMV, respectively, encoding the viral silencing suppressor protein (VSR) of each virus (Table 1). Since then, amiRNAs have been widely used to induce resistance against a large number of DNA and RNA viruses in multiple model and crop species (Table 1). AmiRNAs produced from a plethora of different *MIRNA* precursors in multiple configurations have been designed to target viral RNAs corresponding to VSRs and other key viral proteins (Table 1). However, one important limitation of the amiRNA approach has been the emergence of virus variants (or “escapes”) with mutated target site sequences in plants expressing single amiRNAs targeting single sites [11,12,13,14].

### 2.2. Syn-tasiRNAs

Syn-tasiRNAs are produced in planta by expressing syn-tasiRNA transgenes containing a functional *TAS* precursor in which a subset of the endogenous tasiRNA sequences is substituted by one or several syn-tasiRNA sequences in tandem (Figure 1b) [8,15]. The syn-tasiRNA transgene is transcribed to produce the syn-tasiRNA primary transcript (pri-syn-tasiRNA) which is cleaved by a miRNA/AGO complex. One of the cleaved products is used by RNA-dependent RNA polymerase 6 (RDR6) as a template to synthesize dsRNA that is sequentially processed by DCL4 into phased syn-tasiRNA duplexes in 21-nucleotide registered with the miRNA cleavage site. Importantly, dsRNA is synthesized from 3′ cleavage products originated by miR173/AGO1 cleavage of *TAS1* precursors, and from 5′ cleavage products resulting from miR390/AGO7 cleavage of *TAS3* precursors (Figure 2). In all cases, syn-tasiRNA guide strands are incorporated into AGO1 to target multiple sites in one or multiple viral RNAs (Figure 1b and Figure 2).

Syn-tasiRNAs were first used to confer multiple virus resistance in transgenic Arabidopsis co-expressing three syn-tasiRNAs against *Cucumber mosaic virus* (CMV) and three syn-tasiRNAs against TuMV from a single *TAS3a* precursor [16]. More recently, *TAS1c* precursors were used to express five and six syn-tasiRNAs, respectively, against *Potato spindle tuber viroid* (PSTVd) in *Nicotiana benthamiana* [17] and *Tomato spotted wilt virus* (TSWV) in *N. benthamiana* [18] and *Solanum lycopersicum* (tomato) [14] (Table 1). Importantly, a study comparing the effects of amiRNA and syn-tasiRNA against TSWV in *S. lycopersicum* reported that syn-tasiRNAs induced higher antiviral resistance most likely because of the simultaneous targeting of multiple viral RNAs [14]. As mentioned before [19], sRNAs produced from large gene fragments in MiRNA-Induced Gene Silencing (MIGS) [20] constructs should not be considered as authentic syn-tasiRNAs. Therefore, the antiviral resistance reported when using MIGS constructs [21,22,23] is not considered here.

## 3. Design of Antiviral Art-sRNAs

The efficacy of a particular art-sRNA depends on multiple factors. The degree of base pairing between the art-sRNA and the target RNA is one of them and is considered by the automated design webtools during the art-sRNA design process (see below). However, other factors, such as target site accessibility and stability, are much difficult to predict. For instance, target site accessibility may be limited in those sites with high secondary structures or occupied by an RNA-binding protein. Moreover, upon infection some target sites in viral RNAs may accumulate nucleotide substitutions compatible with viral replication but affecting art-sRNA binding and activity. It is also possible that VSRs interfere with art-sRNA biogenesis or action. For all these reasons, it is difficult to predict if a particular art-sRNA will be effective in vivo. However, we provide next a series of basic rules for art-sRNA design. We also describe the diverse viral regions that have been targeted with art-sRNAs and discuss the strategy of targeting conserved nucleotide sequences in viral RNAs.

### 3.1. General Design Rules

The two main webtools for the automatized design of art-sRNAs are WMD3 (from Web MicroRNA Designer 3) [9] and P-SAMS (from Plant Small RNA
Maker Suite) [24], which were optimized for both the effectiveness and the specificity of the designed art-sRNA. Regarding the effectiveness, art-sRNAs are designed to extensively base pair with the target RNA, with limited or no mismatches near the cleavage site and the 5′ seed region of the art-sRNA, and at least one mismatch in the 3′ end of the art-sRNA to avoid transitivity (due to priming and extension by RDRs). Regarding the specificity, the art-sRNA is designed to target exclusively the intended target(s) with no off-target effects. The specificity of the art-sRNA is assessed through the genome-wide computational analysis of all possible base pairing interactions between the candidate art-sRNA and the complete set of cellular transcripts. Thus, this type of analysis is possible in plant species with an annotated transcriptome or an expressed sequence tag (EST) collection. Other general design criteria are: (i) position 1 of the art-sRNA is a U to favor AGO1 association, (ii) position 19 of the art-sRNA is a C to generate an art-sRNA with a star strand including an AGO1 non-preferred 5′G (in P-SAMS), and (iii) the hybridization energy of the amiRNA/target RNA interaction is between −35 and −40 kcal/mole (in WMD3). The complete set of rules specific to WMD3 and P-SAMS designs have been explained in detail previously [9,24]. For the design of highly specific art-sRNAs against viruses, both webtools allow the input of the target viral RNA sequence in FASTA format and the activation of the target specificity module.

### 3.2. Selection of Target Sequences in Viral RNAs

Art-sRNAs have been used to interfere with key viral functions, such as the suppression of host defense mechanisms by VSRs or the replication of viral RNAs by RNA-dependent RNA polymerases (RdRPs) [47]. VSR RNAs have been frequent targets, as reported for 2b of CMV, AV2 of *Tomato leaf curl New Delhi virus* (ToLCNDV), HC-Pro of *Potato virus Y* (PVY), TuMV and *Wheat streak mosaic virus* (WSMV), NSs of *Tomato spotted wild virus* (TSWV), P19 of *Tomato bush stunt virus* (TBSV), p25 of *Potato virus X* (PVX), P69 of TYMV, Rep and RepA of *Wheat dwarf virus* (WDV), and V2 of *Cotton leaf curl Burewala virus* (CLCuBuV) (Table 1). Other studies have described the targeting of RdRP RNAs of *Cymbidium mosaic virus* (CYmMV), *Green mottle mosaic virus* (CGMMV), *Odontoglossum ringspot virus* (ORSV), TSWV, *Water silver mottle virus* (WSMoV), and WDV, or of coat protein (CP) RNAs of *Cucumber green mottle mosaic virus* (CGMMV), *Grapevine fanleaf virus* (GFLV), *Plum pox virus* (PPV), PVY, *Rice black streaked dwarf virus* (RBSDV), *Rice stripe virus* (RSV), *Tobacco etch virus* (TEV), TuMV, and *Ugandan cassava brown streak virus* (UCBSV) (Table 1). Target sequences in viral RNAs were mostly included in coding regions, but also in 3′ untranslated or antigenomic regions. The complete list of viral regions targeted with art-sRNAs is shown in Table 1.

When selecting the region(s) of the virus to be targeted, a frequent strategy has been the identification of regions with conserved nucleotide sequences. In principle, the targeting of such conserved regions should minimize the possibility of emergence of escape mutants and/or allow the multitargeting of different virus isolates or species. For instance, an amiRNA targeting conserved sequences in the 3′ end of TuMV CP cistron induced high levels of antiviral resistance when stably expressed in Arabidopsis, and no virus variants with mutated target sites were observed [31]. In a different study, an amiRNA designed to target a conserved site of low entropy value included in TSWV RdRP RNAs induced high resistance against two different TSWV isolates when transiently expressed in *N. benthamiana* [18]. Interestingly, when the same amiRNA was stably expressed in tomato plants, TSWV variants with nucleotide substitutions at the conserved target site were observed, most likely as a consequence of the higher selective pressure imposed in amiRNA-overexpressing transgenic plants [14]. These mutations were silent, did not modify the amino acid sequence of the viral RdRP and, therefore, did not affect viral replication [14]. Hence, this study alerts that the targeting of a conserved sequence may limit but not fully impede the emergence of virus escapes.

## 4. Recent Advances in Art-sRNAi for Enhanced Antiviral Resistance

Important improvements in antiviral art-sRNAi methodologies have been reported lately and are presented next. We also discuss the possibility of further improving art-sRNAi based on recent findings in the biogenesis and mode of action of plant miRNAs.

### 4.1. Identification of Effective Art-sRNAs with High Antiviral Activity

As explained above, the antiviral efficacy of a particular art-sRNA is difficult to predict a priori. Thus, systems for the rapid screening of the antiviral activity of large numbers of art-sRNAs are necessary for the identification of effective art-sRNAs prior to the time-consuming generation of stably transformed plants. Recently, a systematic and high-throughput methodology for the simple and fast-forward design, generation, and functional analysis of large numbers of art-sRNA constructs has been described [48]. Briefly, highly specific antiviral amiRNAs are designed with the P-SAMS web tool [24], and selected amiRNA candidate sequences are cloned into *Bsa*I/*ccd*B “B/c” vectors [19], a new generation of vectors for one-step amiRNA cloning and efficient gene silencing in plants [49]. Each amiRNA construct is transiently expressed in several *N. benthamiana* plants, which are subsequently inoculated with the virus of interest. The antiviral activity of each amiRNA construct is assessed by monitoring viral symptom appearance, and through molecular analysis of virus accumulation in plant tissues. This methodology was successfully applied to identify highly effective amiRNAs against RNAs of PSTVd [17] and TSWV [18] in *N. benthamiana*. Other amiRNA screening systems, such as ETPamiR screenings in plant protoplasts [50,51], have been successfully applied to identify highly effective amiRNAs against endogenous genes but not yet against viral RNAs.

Another possibility is to express as art-sRNAs those immunologically effective small interfering RNAs (esiRNAs) that are generated by DCLs from viral dsRNAs produced during viral replication [39]. Because the majority of virus-derived sRNAs are ineffective against the producing virus [52,53,54,55], it has been difficult to distinguish esiRNAs reliably and efficiently until very recently. Gago-Zachert and colleagues recently reported the identification of TBSV-derived esiRNAs using an in vitro system of cytoplasmic extracts from *N. tabacum* BY-2 protoplasts expressing TBSV dsRNAs and selected AGO members [39]. First, 21-nucleotide sRNAs derived from TBSV dsRNAs and loaded by AGO1 or AGO2 were identified by immunoprecipitation followed by sRNA sequencing. Second, the cleavage efficiency of each of these sRNAs was analyzed in in vitro cleavage assays, and third, the protective effects of several of these siRNAs were analyzed in *N. benthamiana* plants by agroinfiltrating each sRNA as an amiRNA, and subsequently inoculating TBSV in the same leaf-sites. Results showed that the functionality of esiRNAs mainly depended on the binding affinity to AGO proteins and the ability to target RNA [39]. Thus, this methodology could be attractive to identify naturally occurring virus-derived sRNAs that are efficient in silencing viral RNAs.

### 4.2. Co-Expression of Multiple Art-sRNAs for Viral RNA Multi-Targeting

In addition to the pioneering work by Niu and colleagues [10], the in vivo co-expression of multiple antiviral amiRNAs through different strategies (Figure 3) has proven effective in several plant/virus pathosystems (Table 1). Furthermore, resistance against PVY was achieved by co-expressing an amiRNA and a siRNA from a *MIRNA* precursor and a short hairpin RNA (shRNA) precursor, respectively, in tandem [37] (Table 1). Despite the success in generating virus resistant plants through all these strategies, methods to generate such amiRNA constructs are rather long and tedious.

On the other hand, *TAS* precursors possess a “natural” multiplexing capability allowing the insertion of multiple syn-tasiRNAs in a single construct (Figure 3). This feature combined with the availability of high-throughput syn-tasiRNA “B/c” vectors [7,49] allows for the efficient generation of antiviral syn-tasiRNA constructs. An interesting possibility is to combine in a single syn-tasiRNA construct a number of effective amiRNAs previously identified in large screens. Indeed, this type of strategy has been already applied and resulted in high levels of resistance against PSTVd and TSWV in *N. benthamiana* and *S. lycopersicum* plants, respectively [14,17]. Interestingly, the comparative analysis of plants expressing a single amiRNA or four syn-tasiRNAs against conserved sites in TSWV RNAs showed that most of the plants expressing syn-tasiRNAs were resistant, while only plants expressing particularly low syn-tasiRNA levels were infected. In contrast, the majority of amiRNA-expressing plants were susceptible, and accumulated virus variants with mutated target sites [14]. These results suggest that the simultaneous multi-targeting of TSWV RNAs with various syn-tasiRNAs most likely limits the ability of the virus to mutate all target sites, whereas subinhibitory amiRNA accumulation favors the emergence of target site mutations in the replicating virus.

### 4.3. Other

Some VSRs counter endogenous miRNA function by interfering with miRNA biogenesis, AGO loading, or AGO/miRNA action [47,56]. For example, it has been reported that TBSV P19 binds to miRNA duplexes [57]. Similarly, Zhang and colleagues recently observed that anti-TBSV amiRNA duplexes were bound and sequestered by P19, and hypothesized that decreasing P19 binding to amiRNA duplexes should reestablish proper duplex processing and amiRNA silencing function [42]. For that purpose, Zhang and colleagues tested if the presence of an asymmetric bulge (AB) in the amiRNA duplex region could affect the interaction between P19 and the amiRNA duplex, as unpaired nucleotides of ABs get flipped out from the RNA helices at the miRNA duplex region according to structural modelling [58,59]. A systematic study of the silencing effect and P19 binding of a series of amiRNA duplexes including an AB at various positions in the guide or star strands showed that in two of the configurations the AB enhanced amiRNA silencing activity and anti-TBSV resistance [42]. Unfortunately, the reasons explaining why ABs at specific positions induced higher interfering effects were not clear. Because this approach can only be used to counteract VSRs that bind amiRNA duplexes, the broad application of this strategy for increased amiRNA-mediated antiviral resistance in plants seems unlikely.

Another recent work reported that plant miRNAs have specific GC signatures required for abundant miRNA production, possibly by influencing the local structure of the precursor to enhance DCL1 partner HYL1-binding and selection [60]. When applying these GC signatures to amiRNAs targeting endogenous or artificial genes, some of the amiRNAs accumulated to higher levels and induced higher target silencing when transiently expressed in *Nicotiana tabacum* leaves [60]. In another recent work, it was suggested that in vivo mRNA structure regulates miRNA cleavage in Arabidopsis [61]. In particular, the single-strandedness of the two nucleotides immediately downstream of the miRNA target site, named Target Adjacent structure Motif (TAM), seems to favor miRNA cleavage [61]. It is tempting to speculate that art-sRNAs targeting viral target sites with TAMs may have an increased cleavage activity and, subsequently, an enhanced antiviral activity. However, whether the targeting of TAM-including target sites or the addition of GC signatures increases the efficacy of antiviral amiRNAs still needs experimental confirmation.

## 5. Application of Art-sRNAs to Control Viral Diseases in Crops

Transgenes producing antiviral art-sRNAs from endogenous sRNA precursors have been introduced in diverse crop species to generate antiviral resistance (Table 1). Regarding amiRNAs, transgenic tomato plants expressing amiRNAs against CMV [28], ToLCNDV [38], or TSWV [14] were resistant, as were barley, maize, rice, and wheat transgenic plants expressing amiRNAs against WDV [40], RBSDV [41], RBSDV/RSV [44], and WSMV [45], respectively. Regarding syn-tasiRNAs, the recent development of highly resistant transgenic tomato plants expressing four different syn-tasiRNAs against TSWV is the only example reported [14]. However, the antiviral resistance of these transgenic crops expressing art-sRNAs has not been examined under field conditions. Indeed, the confirmation that the durability of the antiviral resistance in the field lasts for multiple generations seems necessary before the approval of a transgenic crop for its commercial release. To date, only a few transgenic crops expressing transgenes including short stretches of viral sequences in sense or antisense orientation have been commercially released (for a recent review see [62]). Unfortunately, the current legislations and long paths for the commercialization of a transgenic crop, particularly in those countries where GMO crops are highly regulated, are still barriers to overcome before the release of the first art-sRNA-based crop onto the market.

An alternative to the transgenic expression of antiviral RNAs is their topical delivery into plants. This approach was first used to interfere with *Alfalfa mosaic virus* (AMV), TEV, or *Pepper mild mottle virus* (PMMoV) infection by mechanically co-inoculating *N. tabacum* leaves with naked dsRNAs of virus sequence and their corresponding target virus [63]. Later, the same strategy was also applied successfully to interfere with the infection of several viroids [64]. Since these early works, dsRNAs of viral sequence have been delivered to plants though diverse methods to induce resistance against a large number of plant viruses in multiple model and crop species (reviewed recently in [65,66,67,68]). However, several limitations of these approaches may include the lack of affordable methods for dsRNA production, and the low specificity, potential toxicity, and reduced efficiency of certain dsRNAs. To date, the exogenous application of art-sRNAs to plants has not been reported. In principle, art-sRNA precursors topically delivered into plants will be processed by the endogenous RNAi machinery to produce the antiviral art-sRNAs. Certainly, some of the limitations of the dsRNA approach, such as the lack of efficient production and delivery methods, may also compromise the successful exogenous application of art-sRNA precursors into plants. In this context, several bacterial systems for the efficient production of recombinant RNAs [69,70,71,72] may be used to produce large amounts of art-sRNA precursors in a time- and cost-effective manner. In addition, a possibility to increase the stability and efficient delivery of art-sRNA precursors could be their conjugation to cationic nanoparticles, clay nanosheets, surfactants, or peptide-based RNA delivery systems, as described for other RNAs [66]. For example, sprayed dsRNAs bound to layered double hydroxide (LDH) nanosheets have been successfully used to confer resistance to PMMoV and CMV in *N. tabacum* [73], in a non-toxic and sustainable manner, and extend the durability of the protection described in previous studies [63]. Thus, the exogenous application of art-sRNA precursors conjugated to new generation nanoparticles may represent a novel, highly efficient, and sustainable strategy to induce antiviral resistance in crops in a GMO-free manner.

## 6. Concluding Remarks and Future Perspectives

Art-sRNAi tools have been broadly used in plants to confer antiviral resistance against multiple RNA and DNA viruses, and to viroids as well. Currently, the relative simplicity of the webtool-assisted design of highly specific antiviral art-sRNA, combined with the availability of efficient cloning methods, facilitates the design and generation of antiviral art-sRNA constructs for plant delivery. However, one important drawback in the use of art-sRNAi is the difficulty to predict the effectiveness of a particular art-sRNA. Recently described high-throughput systems for rapid in vitro or in vivo screening of the antiviral activity of virus-derived sRNAs or computationally designed art-sRNAs, respectively, seem to have overcome this limitation. Another drawback is the emergence of resistance-breaking virus variants with mutated target sites when using single amiRNAs targeting single sites in viral RNAs. In this case, the artificial multiplexing of amiRNAs in different precursor configurations or the use of syn-tasiRNA precursors, both allowing the co-expression of multiple art-sRNAs, should circumvent this problem. The synchronized targeting of multiple viral RNAs by co-expressed art-sRNAs may minimize the possibility that the virus simultaneously mutates all different target sites to fully escape each art-sRNA, and thus enhance the antiviral resistance.

In the current genome editing era of bacterial CRISPR/Cas-based technologies, we anticipate that art-sRNAi tools will continue to be broadly used to confer antiviral resistance in plants because of their unique features of high simplicity, specificity, and efficacy, as well as for their multiplexing capability and for the availability of high-throughput methodologies for the design, generation, and validation of art-sRNAi constructs. The development of efficient methodologies for the production and topical delivery to plants of art-sRNA precursors, as well as a better knowledge of the basic mechanisms governing art-sRNA biogenesis, mode of action, and viral targeting, are needed to further refine art-sRNAi tools in view of their broader use for enhanced crop protection.

## Figures and Tables

**Figure 1 plants-09-00669-f001:**
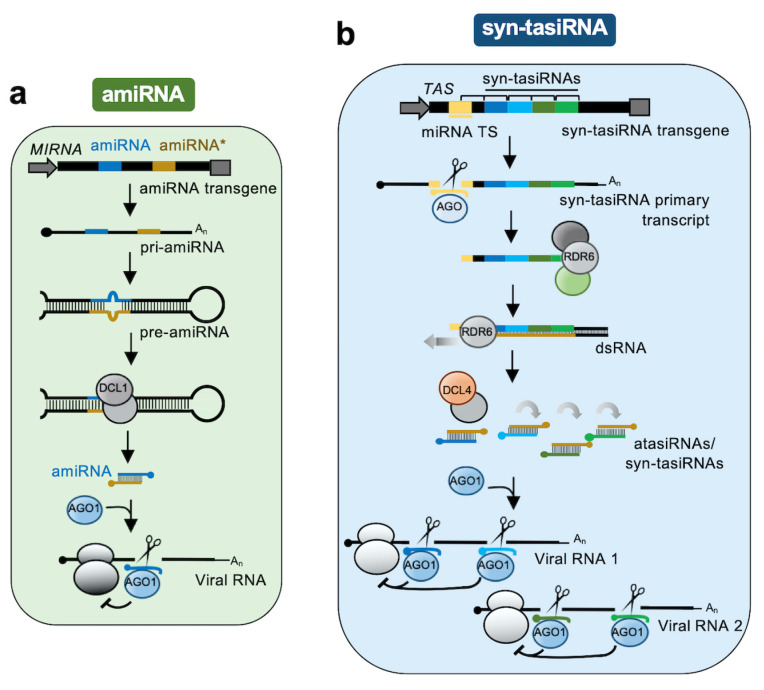
Antiviral art-sRNA pathways in plants. (**a**) The antiviral amiRNA pathway. The amiRNA transgene expresses a monocistronic *MIRNA* precursor sequentially processed into an amiRNA targeting a single site in a single viral RNA. (**b**) The antiviral syn-tasiRNA pathway. The syn-tasiRNA transgene expresses a polycistronic *TAS* precursor sequentially processed into four different syn-tasiRNAs targeting multiple sites in multiple viral RNAs. Both amiRNA and syn-tasiRNA guide strands associated with AGO1 to silence viral RNAs through endonucleolytic cleavage or translational inhibition.

**Figure 2 plants-09-00669-f002:**
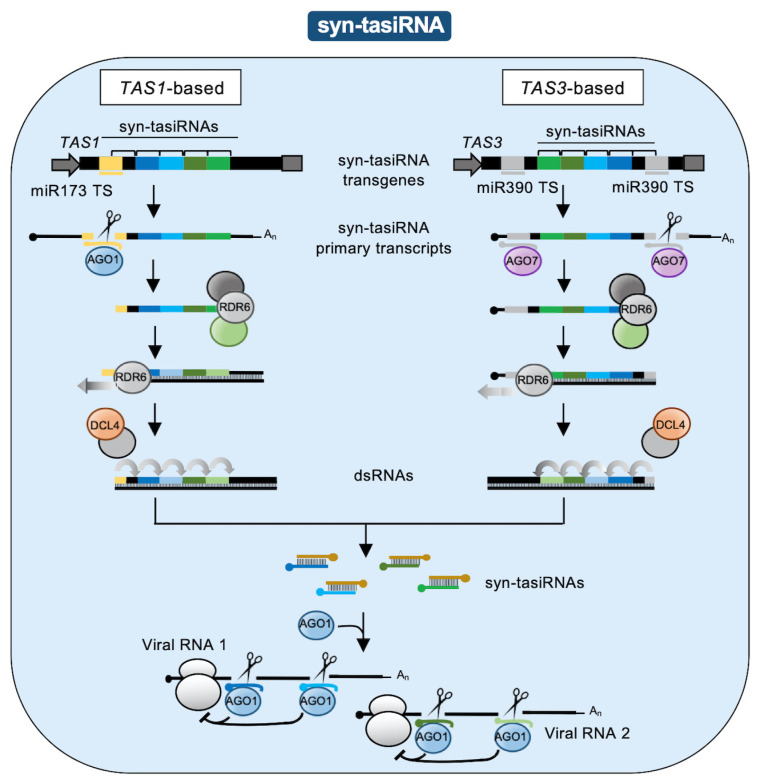
*TAS*-based antiviral syn-tasiRNA pathways in plants. Left, *TAS1*-based syn-tasiRNA pathway. Right, *TAS3*-based syn-tasiRNA pathway. Other details are as in Figure 1.

**Figure 3 plants-09-00669-f003:**
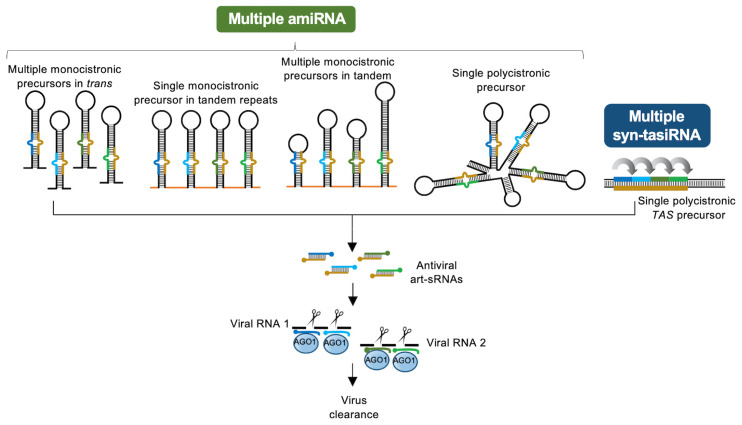
Strategies for the co-expression of multiple art-sRNAs in plants. Multiple amiRNAs are generated from several monocistronic precursors expressed in trans, from a single monocistronic precursor in tandem repeats, from multiple monocistronic precursors in tandem or from a single polycistronic precursor. Multiple syn-tasiRNAs are generated from a single polycistronic *TAS* precursor. Other details are as in Figure 1.

**Table 1 plants-09-00669-t001:** Uses of art-sRNAi to induce antiviral resistance in plants.

Art-sRNA Class/Type of Precursor	Precursor(s)	No. Art-sRNAs	Target Virus(es) ^1^	Target Region(s) ^2^	Effects	Type of Expression ^3^/Transformation Method	Plant Species	Ref.
amiRNA/single monocistronic	*AthMIR156*	1	CGMMV	CP	High protection.	TE/leaf agroinfiltration	*Nicotiana benthamiana*	[25]
	*AthMIR159a*	1	CBSV	P1, P3, NIb, 3′UTR	High protection when targeting P1 or. NIb (also against UCBSV).	SE/leaf disc	*Nicotiana benthamiana*	[26]
		1	CMV	3′ UTR (RNA3)	High protection when target site is not in tRNA-like structures.	SE/floral dip	*Arabidopsis thaliana*	[27]
		1	CMV	3′ UTR (RNA3)	High protection when target site is not in tRNA-like structures.	SE/leaf disc	*Nicotiana tabacum*	[27]
		1	CMV	2a/2b, 3′ UTR	Higher protection when targeting 2a/2b.	SE/cotyledonary explants	*Solanum lycopersicum*	[28]
		1	PVX	p25	High protection, even at increased viral pressure.	SE/leaf disc	*Nicotiana tabacum*	[29]
		1	PVY	HC-Pro	High protection, even at increased viral pressure.	SE/leaf disc	*Nicotiana tabacum*	[29]
		1	TSWV	N, NSs	High protection when targeting N. No protection when targeting NSs.	TE/leaf agroinfiltration	*Nicotiana benthamiana*	[30]
						SE/leaf disc	*Nicotiana tabacum*	
		1	TuMV	HC-Pro	High protection.	SE/floral dip	*Arabidopsis thaliana*	[10]
					Virus escapes emerge at subinhibitory amiRNA concentrations.	SE/floral dip	*Arabidopsis thaliana*	[13]
					Intermediate protection.	SE/floral dip	*Arabidopsis thaliana*	[31]
		1	TuMV	CP	High protection.	SE/floral dip	*Arabidopsis thaliana*	[31]
		1	TYMV	P69	High protection.	SE/floral dip	*Arabidopsis thaliana*	[10]
		1	UCBSV	P1, P3, CI, NIb, CP, 3′UTR	High protection when targeting P1 or. CP (lower against UCBSV).	SE/leaf disc	*Nicotiana benthamiana*	[26]
		1	WSMoV	A, B1, B2, C, D, E (RdRP)	Intermediate protection when targeting B2 and D.	SE/leaf disc	*Nicotiana benthamiana*	[32]
	*AthMIR164*	1	CGMMV	MP	High protection.	TE/leaf agroinfiltration	*Nicotiana benthamiana*	[25]
	*AthMIR167b*	1	PVX	p25	Intermediate protection. Broken resistance after re-inoculation.	SE/leaf disc	*Nicotiana tabacum*	[29]
		1	PVY	HC-Pro				
	*AthMIR169a*	1	CLCuBuV	V2	Low or high protection when the precursor was or was not modified, respectively.	SE/leaf disc	*Nicotiana benthamiana*	[33]
	*AthMIR171a*	1	CMV	2b	Inhibition of 2b silencing suppressor function.	TE/leaf agroinfiltration	*Nicotiana benthamiana*,	[34]
					63.3% of the lines were resistant.	SE/leaf disc	*Nicotiana tabacum*	
		1	CGMMV	Rep	High protection.	TE/leaf agroinfiltration	*Nicotiana benthamiana*	[25]
		1	PVX	p25	Intermediate protection. Broken resistance after re-inoculation.	SE/leaf disc	*Nicotiana tabacum*	[29]
		1	PVY	HC-Pro				
	*AthMIR319a*	1	GFLV	CP	AmiRNAs are active against GFLV target sites located in a GUS mRNA sensor.	TE/somatic embryos at cotyledonary stage	*Vitis vinifera*	[35]
		1	PVY	CI, NIa, NIb, CP	Higher protection when targeting NIb or CP.	SE/leaf disc	*Nicotiana tabacum*	[36]
		1	PVY^O^ + PVY^N^	NIb (PVY^O^) + NIb (PVY^N^)	33% and 17% of the lines were resistant to PVY^O^ and PVY^N^, respectively.	SE/leaf disc	*Nicotiana tabacum*	[37]
		1	TEV	CI, NIa, NIb, CP	Higher protection when targeting NIb or CP.	SE/leaf disc	*Nicotiana tabacum*	[36]
		1	ToLCNDV	AV1, AV1 + AV2	High tolerance when targeting AV1 + AV2. Moderate tolerance when targeting AV1.	SE/cotyledonary explants	*Solanum lycopersicum*	[38]
	*AthMIR390a*	1	PSTVd	TL, C, V [PSTVd(+)],	Delay of viroid accumulation in all cases.	TE/leaf agroinfiltration	*Nicotiana benthamiana*	[17]
		1	PSTVd	TL, P, C, V, TR [PSTVd(−)]	Delay of viroid accumulation when targeting TL and C.			
		1	TBSV	5′ terminus TBSV(+) RNA	40–90% of plants were symptom-free.	TE/leaf agroinfiltration	*Nicotiana benthamiana*	[39]
		1	TSWV	N, NSm, NSs, RdRP	50–100% of the plants did not accumulate TSWV when targeting NSm or RdRP.	TE/leaf agroinfiltration	*Nicotiana benthamiana*	[18]
		1	TSWV	RdRP	22% of the lines were resistant.	SE/cotyledonary explants	*Solanum lycopersicum*	[14]
	*HvuMIR171a*	1	WDV	MP, Rep, RepA, RepA + Rep	AmiRNAs against Rep and RepA + Rep were selected based on a reporter system.	TE/leaf agroinfiltration	*Nicotiana benthamiana*	[40]
	*SlyMIR159a*	1	ToLCNDV	AV1, AV1 + AV2	High tolerance when targeting AV1 + AV2. Moderate tolerance when targeting AV1.	SE/cotyledonary explants	*Solanum lycopersicum*	[38]
	*SlyMIR168a*	1	ToLCNDV	AV1, AV1 + AV2	High tolerance when targeting AV1 + AV2. No accumulation of amiRNAs against AV1.			
	*ZmaMIR159a*	1	RBSDV	P6	High protection.	SE/ear immature embryos	*Zea mays*	[41]
amiRNA/single monocistronic in tandem repeats	*AthMIR159a*	2	PVX + PVY	P25 (PVX) + HC-Pro (PVY)	High protection against both viruses.	SE/leaf disc	*Nicotiana tabacum*	[29]
			TuMV + TYMV	HC-Pro (TuMV) + P69 (TYMV)	High protection against both viruses	SE/floral dip	*Arabidopsis thaliana*	[10]
		3	WSMoV	RdRP	High protection	SE/leaf disc	*Nicotiana benthamiana*	[32]
	*AthMIR171a*	2	TBSV	P19 + P33	Effective antiviral silencing in agroinfiltrated leaves.	TE/leaf agroinfiltration	*Nicotiana benthamiana*	[42]
	*AthMIR319a*	2	PVY^O^ + PVY^N^	NIb (PVY^O^) + NIb (PVY^N^)	52% and 30% of the lines were resistant to PVY^O^ and PVY^N^, respectively.	SE/leaf disc	*Nicotiana tabacum*	[37]
	*HvuMIR171a*	3	WDV	Rep + RepA	One line was fully resistant.	SE/spike immature embryos	*Hordeum vulgare*,	[40]
					Efficient silencing of the overexpressed Rep mRNA at 15ºC and 23ºC.	TE/leaf agroinfiltration	*Nicotiana benthamiana*	
	*OsaMIR528*	2	CymMV + ORSV	RdRP (CymMV) + RdRP (ORSV)	73% and 16% of the lines were resistant to CymMV and ORSV, respectively.	SE/leaf disc	*Nicotiana benthamiana*	[43]
	*OsaMIR528*	2	RBSDV + RSV	CP (RBSDV) + CP (RSV)	54% and 27% of the lines were resistant to RBSDV and RSV, respectively.	SE/scutellum-derived calli	*Oryza sativa*	[44]
amiRNA/single polycistronic	*OsaMIR395*	5	WSMV	5′ UTR + P1 + HC-Pro + P3	Three types of lines were observed: completely immune; initially resistant with resistance breaking down over time; and initially susceptible followed by plant recovery.	SE/microparticle bombardment of embryos	*Triticum aestivum*	[45]
amiRNA/multiple monocistronic in tandem	*AthMIR157* + *AthMIR159* + *AthMIR171*	3	PPV	CP	No protection.	SE/hypocotyl slices	*Prunus domestica*	[46]
amiRNA/multiple monocistronic in *trans*	*AthMIR159a*	2	TuMV	CP + HC-Pro	High protection.	SE/floral dip	*Arabidopsis thaliana*	[31]
	*AthMIR390a*	7	TBSV	5′ terminus TBSV(+) RNA	80% of plants were symptom-free.	TE/leaf agroinfiltration	*Nicotiana benthamiana*	[39]
amiRNA + siRNA/multiple monocistronic in tandem	*AthMIR319a +* shRNA	2	PVY^O^ + PVY^N^	NIb (PVY^O^) + NIb (PVY^N^)	69% and 47% of the lines were resistant to PVY^O^ and PVY^N^, respectively.	SE/leaf disc	*Nicotiana tabacum*	[37]
syn-tasiRNA/single polycistronic	*AthTAS1c*	1	TSWV	RdRP	Delay of viroid accumulation.	TE/leaf agroinfiltration	*Nicotiana benthamiana*	[7]
		4	TSWV	NSm + RdRP	100% of the plants were resistant.	TE/leaf agroinfiltration	*Nicotiana benthamiana*,	[18]
					83% of the lines were resistant.	SE/cotyledonary explants	*Solanum lycopersicum*	[14]
		5	PSTVd	TL + C + V + TR [PSTVd(+)] + TL + TR [PSTVd(−)]	Delay of viroid accumulation.	TE/leaf agroinfiltration	*Nicotiana benthamiana*	[17]
	*AthTAS3a*	6	CMV + TuMV	RdRP + 2b (CMV) + P1 + P3 + CP (TuMV)	All lines were resistant to both viruses.	SE/floral dip	*Arabidopsis thaliana*	[16]

^1^ CBSV, Cassava brown streak virus; CGMMV, Cucumber green mottle mosaic virus; CLCuBuV, Cotton leaf curl Burewala virus; CMV, Cucumber mosaic virus; CymMV, Cymbidium mosaic virus; GFLV, Grapevine fanleaf virus; ORSV, Odontoglossum ringspot virus; PPV, Plum pox virus; PSTVd, Potato spindle tuber viroid; PVX, Potato virus X; PVY, Potato virus Y; PVY^N^, Potato virus Y nectrotic strain; PVY^O^, Potato virus Y common strain; RBSDV, Rice black streaked dwarf virus; RSV, Rice stripe virus; TBSV, Tomato bush stunt virus; TEV, Tobacco etch virus; ToLCNDV, Tomato leaf curl New Delhi virus; TSWV, Tomato spotted wilt virus; TuMV, Turnip mosaic virus; TYMV, Turnip yellow mosaic virus; UCBSV, Ugandan cassava brown streak virus; WDV, Wheat dwarf virus; WSMoV, Watermelon silver mottle virus; WSMV, Wheat streak mosaic virus. ^2^ C, conserved domain; CI, cylindrical inclusion protein; CP, coat protein; HC-Pro, helper component proteinase; MP, movement protein; N, nucleocapsid protein; NSs, nucleocapsid segment S silencing suppressor protein; NSm, nucleocapsid segment S movement protein; NIa, nuclear inclusion a protein; NIb, nuclear inclusion b protein; P, pathogenic domain; Rep and RepA, proteins associated with viral replication; RdRP, RNA-dependent RNA polymerase; TL, terminal left domain; TLS, tRNA-like structure; TR, terminal right domain; UTR, untranslated region; V, variable domain.^3^ SE, stable expression; TE, transient expression.

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
