# Peer review of "Artificial Small RNA-Based Silencing Tools for Antiviral Resistance in Plants"

_plants, 2020, doi:10.3390/plants9060669_

Round 1
Reviewer 1 Report
Review report about Cisneros and Carbonell: „ Artificial small RNA-based silencing tools for antiviral resistance in plants” manuscript.
Virus infection can cause serious crop damage worldwide, and will have an even bigger importance in the future. As traditional plant breeding needs long time new, biotechnology-based methods for producing virus resistant plants emerged. In these transgenic plants only small RNAs are produced, why their usage will or can be acceptable even for the most anti GM people. Although usage of these new techniques is on its way now, it can be revised and updated as new cutting-edge techniques become available.
In this review we get a very impressive outlook about the stage, the tricks and possible emerging problems of these new techniques. The review not only describe in a very reader-friendly way the biology behind art-sRNA based methods, but contains a comprehensive database about the targeted viruses and plants.
I think that this paper is very well written and has a wide interest, why I think it can be accepted for publication in MDPI Plants even with further corrections.
Author Response
We want to thank the reviewer for the enthusiastic and positive comments about our manuscript.
Reviewer 2 Report
Manuscript ID: plants-808524
Type of manuscript: Review
Title: Artificial Small RNA-Based Silencing Tools for Antiviral Resistance in Plants
Using silencing tools for plant antiviral protection based on artificial small RNAs is a promising approach. Thus, the topic of this Ms is relevant for Plants. This manuscript reviews the recent important information on the features of the tools based on small artificial RNA, provides the information on design of antiviral RNAs, and describes the recent advances in identification of effective small RNAs for plant antiviral resistance. There are several important issues in this manuscript where improvements could contribute to the completeness and comprehensibility of the review so that it would make an essential contribution to the field of molecular plant protection (extensive minor revision).
Specific comments:
- The structure and content of the Table 1 should be improved. There is too much empty space in two first columns, so the space is not efficiently used. It appears that the first column should be deleted or moved. For example, delete this column and instead insert a horizontal line with art-sRNA class for the other columns before each art-sRNA class begins. In addition, cale down the volume of second column or modify it in a different way to optimize space.
- I recommend that two new columns should be included in the Table 1 to make it more informative. For example, “Effect” column (some details on the antiviral effect, e.g. its strength and maybe time of its maintenance). The second column – the means of small RNA introduction in plants, i.e. Agroinfiltration of constructs in leaves, floral-dip Arabidopsis transformation, agrobacterium-mediated transformation of cotyledonary explants etc. Maybe also include the used plasmid vector name and/or promoter type.
- It is not completely clear for me when and where the term and abbreviation “art-sRNA” was coined. Maybe it would be useful to include this information and a citation in the introduction.
- Please compare the efficiency of amiRNAs and syn-tasiRNAs for plant protection against plant viruses. For example, at page 4, line 108.
- Include a list of all most important used abbreviations at the end of this manuscript.
- Expression of transgenes encoding small artificial RNAs in plants is the most developed approach of small RNA-based plant antiviral resistance. How about application of this technique to field conditions for plant antiviral resistance? Please discuss this. There are discussions around transgenic plants. In addition, discuss other present and/or potential means to introduce small artificial RNAs in plants for plant resistance, e.g. exogenous application of small RNAs onto plant surfaces, maybe something else? Include the references and include more discussion on the perspectives of artificial RNAs for plant antiviral resistance, especially for field conditions.
How about dsRNA application? For example, there is your paper Carbonell “Double-stranded RNA interferes in a sequence-specific manner with the infection of representative members of the two viroid families”. Virology 371 (2008) 44 – 53. Please also include some other citations.
- At the same time, there are too many self-citations in the manuscript. Please check all of them (whether they are indeed necessary).
- page 2, line 67-68. What * means here?
Moderate English corrections are required.
Some examples:
- Abstract, lines 17-18. “Still, the use of art-sRNAs as an antiviral tool has limitations such as the difficulty to predict the efficacy of a particular art-sRNA, or the emergence of virus variants with mutated target sites escaping to art-sRNA-mediated degradation.” Correct to “Application of art-sRNAs as an antiviral tool has limitations, such as the difficulty to predict the efficacy of a particular art-sRNA or the emergence of virus variants with mutated target sites escaping to art-sRNA-mediated degradation.”
I think that “Still,…” should not be used at the beginning of a sentence.
Please also separate by commas “…such as..” parts of a sentence throughout the manuscript. For example, page 1, lines 37-38.
- Abstract, line 19. Correct “Here we review..” to “Here, we review..”.
Page 2, line 45. “Here, we describe”. Correct similar mistakes in the Ms.
- page 3, line 78. Correct “Since,…” to “Since then,…”. Correct this kind of mistakes.
- page 4, line 124. Correct “…, which optimize…” to “….., which were optimized”
- line 152. “Art-sRNAs has sought to …” English should be corrected here.
- line 174. Correct “when this same” to “when the same”
Author Response
We would like to thank the reviewer for the thorough and insightful review of our manuscript that has contributed to increase its impact and clarity. We next provide point by point answers to each of the reviewer’s comments (R=Reviewer; A=Authors).
R: Using silencing tools for plant antiviral protection based on artificial small RNAs is a promising approach. Thus, the topic of this Ms is relevant for Plants. This manuscript reviews the recent important information on the features of the tools based on small artificial RNA, provides the information on design of antiviral RNAs, and describes the recent advances in identification of effective small RNAs for plant antiviral resistance. There are several important issues in this manuscript where improvements could contribute to the completeness and comprehensibility of the review so that it would make an essential contribution to the field of molecular plant protection (extensive minor revision).
A: We value that the reviewer appreciates the relevance of the topic reviewed here, and we appreciate the general positive comments about our manuscript. We also agree that the manuscript can be improved as suggested.
R: Specific comments:
- The structure and content of the Table 1 should be improved. There is too much empty space in two first columns, so the space is not efficiently used. It appears that the first column should be deleted or moved. For example, delete this column and instead insert a horizontal line with art-sRNA class for the other columns before each art-sRNA class begins. In addition, cale down the volume of second column or modify it in a different way to optimize space.
A: We totally agree that there is too much empty space in the two first columns. We followed the reviewer’s suggestions and deleted the first row and reduced the volume of the second column. Actually, we merged the first and second column into a new first column titled “art-sRNA class/type of precursor” containing the information from the previous first two columns. We really thank the reviewer for the suggestion as now the amount of empty space has been reduced considerably.
R: I recommend that two new columns should be included in the Table 1 to make it more informative. For example, “Effect” column (some details on the antiviral effect, e.g. its strength and maybe time of its maintenance). The second column – the means of small RNA introduction in plants, i.e. Agroinfiltration of constructs in leaves, floral-dip Arabidopsis transformation, agrobacterium-mediated transformation of cotyledonary explants etc. Maybe also include the used plasmid vector name and/or promoter type.
A: As suggested, two new columns titled “Effect” and “Type of expression/transformation method” were included in Table 1. We totally agree that with the new added information Table 1 is now more informative, and thus truly thank the reviewer for the nice suggestion.
R: It is not completely clear for me when and where the term and abbreviation “art-sRNA” was coined. Maybe it would be useful to include this information and a citation in the introduction.
A: The abbreviation “art-sRNA” was first used in the López-Dolz et al. NAR 2020 paper. As per reviewer’s suggestion, a reference to this paper was included in the introduction section when “art-sRNA” is first used.
R: Please compare the efficiency of amiRNAs and syn-tasiRNAs for plant protection against plant viruses. For example, at page 4, line 108.
A: The text “Importantly, a study comparing the effects of amiRNA and syn-tasiRNA against TSWV in S. lycopersicum reported that syn-tasiRNAs induced higher antiviral resistance most likely because of the simultaneous targeting of multiple viral RNAs [14].” Was added at page 4 line 108. We also note that the efficiency of amiRNAs and syn-tasiRNA was already compared in the text in the paragraph “Interestingly, the comparative analysis of plants expressing a single amiRNA or four syn-tasiRNAs against conserved sites in TSWV RNAs showed that most of the plants expressing syn-tasiRNAs were resistant, while only plants expressing particularly low syn-tasiRNA levels were infected. In contrast, the majority of amiRNA-expressing plants were susceptible, and accumulated virus variants with mutated target sites [14]. These results suggest that the simultaneous multi-targeting of TSWV RNAs with various syn-tasiRNAs most likely limits the ability of the virus to mutate all target sites, whereas subinhibitory amiRNA accumulation favors the emergence of target site mutations in the replicating virus.” at the end of section 4.2.
R: Include a list of all most important used abbreviations at the end of this manuscript.
A: As per reviewer’s suggestion, a list of all most important used abbreviations was included at the end of the manuscript.
R: Expression of transgenes encoding small artificial RNAs in plants is the most developed approach of small RNA-based plant antiviral resistance. How about application of this technique to field conditions for plant antiviral resistance? Please discuss this. There are discussions around transgenic plants. In addition, discuss other present and/or potential means to introduce small artificial RNAs in plants for plant resistance, e.g. exogenous application of small RNAs onto plant surfaces, maybe something else? Include the references and include more discussion on the perspectives of artificial RNAs for plant antiviral resistance, especially for field conditions.
A: We appreciate and share the reviewer’s concern about discussing art-sRNA application methods to field conditions. A new section #5 titled “Application of art-sRNAs to control viral diseases in crops”, was included in the manuscript for this purpose. Briefly, in this new section we discuss the different art-sRNAi transgenic crops developed for antiviral resistance, and their potential commercial release. We also speculate on the possibility of applying art-sRNA precursors in a GMO-free manner, through their exogenous application (e.g. conjugated to nanoparticles) as reported for antiviral dsRNAs.
R: How about dsRNA application? For example, there is your paper Carbonell “Double-stranded RNA interferes in a sequence-specific manner with the infection of representative members of the two viroid families”. Virology 371 (2008) 44 – 53. Please also include some other citations.
A: As stated in the previous response, dsRNA application was discussed in the new section 5 with appropriate citations.
R: At the same time, there are too many self-citations in the manuscript. Please check all of them (whether they are indeed necessary).
A: As per reviewer’s suggestion, we deleted old references #48 and #51 as we considered them not absolutely necessary.
R: page 2, line 67-68. What * means here?
A: * refers to the miRNA star strand, which is the strand of the duplex besides the guide strand. We replaced “*” by “star” in the main text. In particular, the text “miRNA/miRNA*” was replaced with the text “miRNA guide/miRNA star”.
R: Moderate English corrections are required.
Some examples:
- Abstract, lines 17-18. “Still, the use of art-sRNAs as an antiviral tool has limitations such as the difficulty to predict the efficacy of a particular art-sRNA, or the emergence of virus variants with mutated target sites escaping to art-sRNA-mediated degradation.” Correct to “Application of art-sRNAs as an antiviral tool has limitations, such as the difficulty to predict the efficacy of a particular art-sRNA or the emergence of virus variants with mutated target sites escaping to art-sRNA-mediated degradation.”
I think that “Still,…” should not be used at the beginning of a sentence.
A: The word “Still” was replaced with “However” to begin this sentence, and also throughout the manuscript. The rest of the sentence was corrected as suggested.
R: Please also separate by commas “…such as..” parts of a sentence throughout the manuscript. For example, page 1, lines 37-38.
A: As suggested, we added commas throughout the manuscript to separate parts of a sentence containing “such as”.
R: Abstract, line 19. Correct “Here we review..” to “Here, we review..”.
A: As suggested, the text “Here we review” was replaced with the text “Here, we review”.
R: Page 2, line 45. “Here, we describe”. Correct similar mistakes in the Ms.
A: We corrected to “Here, we describe” in page 2 line 45, and also corrected similar mistakes throughout the manuscript.
R: page 3, line 78. Correct “Since,…” to “Since then,…”. Correct this kind of mistakes.
A: “Since” was replaced with “Since then” in page 3 line 78. We corrected this kind of mistake thoughout the manuscript.
R: page 4, line 124. Correct “…, which optimize…” to “….., which were optimized”
A: ·…, which optimized…” was replaced with “…, which were optimized”.
R: line 152. “Art-sRNAs has sought to …” English should be corrected here.
A: “Art-sRNAi has sough to” was replaced with “Art-sRNAs have been used”.
R: line 174. Correct “when this same” to “when the same”
A: “When this same” was replaced with “when the same”. In line 174.
